# Experiences of Underground Mine Backfilling Using Mine Tailings Developed in the Andean Region of Peru: A Green Mining Solution to Reduce Socio-Environmental Impacts

Carlos Cacciuttolo [1,*] and Alex Marinovic [2]

1   Civil Works and Geology Department, Catholic University of Temuco, Temuco 4780000, Chile
2   Facultad de Ingeniería, Universidad Privada del Norte, Cajamarca 06001, Peru; alex.marinovic@upn.edu.pe
*   Correspondence: ccacciuttolo@uct.cl or carlos.cacciuttolo@gmail.com

**Abstract:** In Peru, socio-environmental conflicts related to the development of mining-metallurgical processes and the responsible disposal of mine tailings have become central issues for accepting mining projects, especially regarding building relationships of trust with the communities. This condition has prompted the Peruvian mining industry to advance in managing alternatives to the conventional surface disposal of mine tailings. A promising and increasingly popular management strategy for mine tailings in Peru is their disposal inside underground mines. This article presents: site-specific conditions, advantages/disadvantages, and lessons learned from practical experiences of mine tailings disposal in underground mines in Peru. In addition, some techniques are highlighted, such as (i) hydraulic fill, (ii) cemented hydraulic fill, and (iii) cemented paste backfill. Finally, this article concludes that the responsible disposal of mine tailings in underground mines is a green mining solution that reduces negative socio-environmental impacts, limiting the generation of acid rock drainage (ARD) and the leaching of metals due to the decrease in contact with oxygen and rainfall, thus mitigating the contamination of surface and underground waters, reducing the footprint of affectation in the territory, and eliminating the emission of particulate matter in the environment.

**Keywords:** mine tailings; underground mine backfilling; hydraulic fill; cemented hydraulic fill; cemented paste backfill; acid rock drainage; metal leaching; sustainability; responsible mining

## 1. Introduction

### 1.1. Alternative Solutions to Dispose of Mine Tailings to Improve Sustainability of Mining

In Peru and the rest of the world, the limitations on the supply of water for mining processes and the responsible disposal of mine tailings have become central aspects of the sustainability of mining activities. These issues are crucial for obtaining the social and environmental license to operate mines in the territory [1–5].

The serious dam failure accidents of mine tailings storage facilities that have occurred in recent years in Canada, Brazil, South Africa, and Tanzania have severely impacted the environment and communities, even in some cases registering deaths of human beings and serious environmental damage to ecosystems [6–9]. This has generated concern from both communities and local authorities about the safety provided by conventional tailings storage facilities that operate on the surface through the construction of large dams [10–12]. It is also in this way that different international organizations involved in mining investment projects, such as ICMM, UN, and PRI, are promoting a global standard for mine tailings management called GISTM in order to improve the governance of these facilities and reduce the risks of failures or mine tailings spills [13–15].

Indeed, the need to reduce the socio-environmental impacts of mine tailings and to build a relationship of trust with the affected communities have translated into a critical need to reduce water consumption in mining processes and an increase in the requirements for the safe and controlled disposal of mine tailings. These issues must be reflected in

compliance with high standards of design, construction, management, and closure of mine tailings storage facilities, which are currently mandatory in Peru [16,17].

This condition has prompted the mining industry to advance in the management of alternatives to the conventional disposal of tailings (CTD) on the surface that contain large amounts of water in the storage reservoir [18]. Some of these alternatives are the so-called best available technologies (BATs), including: (i) thickened high-density tailings (TTD), (ii) paste tailings (PTD), and (iii) filtered tailings (FTD) [19]. These technologies allow great savings in water consumption, the mitigation of the environmental impacts associated with its disposal, and the reduction of the construction of large storage dams [20]. However, they also require significant investments in both capital costs and operating costs, but in the long term, they are profitable in terms of Environmental, Social, and Governance (ESG), social responsibility, and sustainability [21].

In Peru, a promising and increasingly popular alternative strategy to surface management of mine tailings is their disposal as hydraulic fills (HF), cemented hydraulic fills (CHF), and cemented paste backfill (CPB) within underground mines. In Peru, there are historical records that detail the beginnings of the application of hydraulic filling in an underground mine in 1937 in the Cerro de Pasco mine. Since that date, the underground mine backfill technique with mine tailings has evolved and gained much experience considering the site-specific local conditions typical of the Andean region of Peru. Many underground mines around the world use different backfill technologies such as rockfill [22], hydraulic fill [23], and paste backfill [24]. It was the presence of mine tailings near the underground mines and their danger to the environment that contributed to the development of backfilling technologies in the underground mines of Peru. Figure 1 shows typical adit access to an underground mine in the Andean region of Peru.

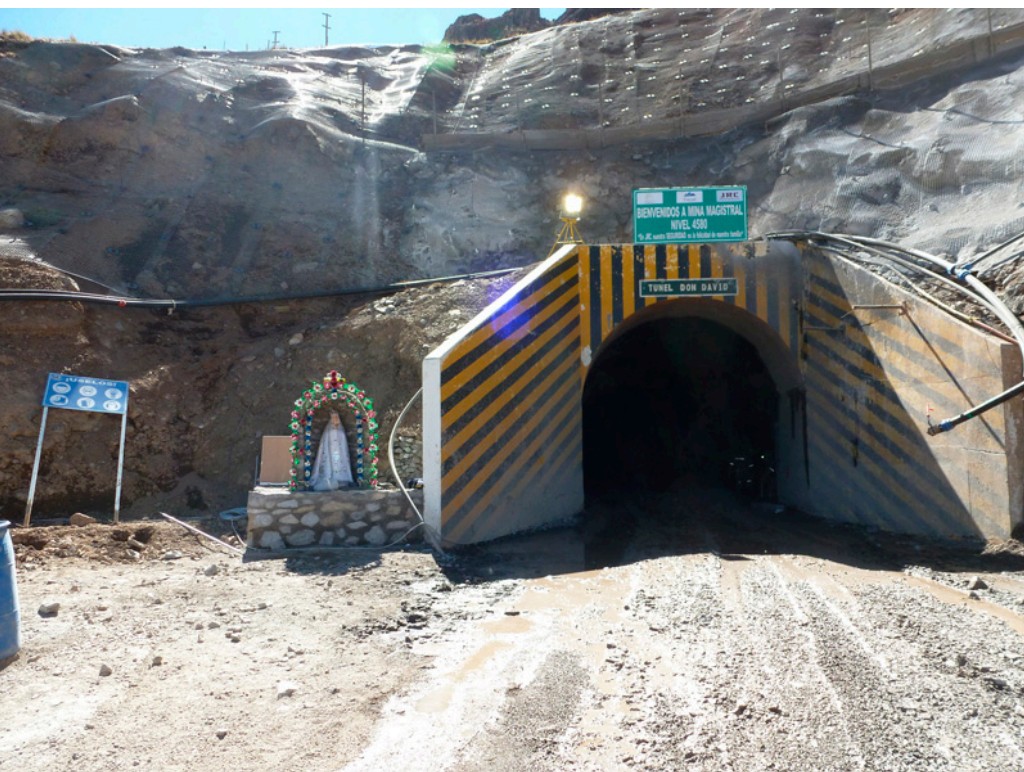

**Figure 1.** Typical adit access to an underground mine in the Andean region of Peru.

The disposal of mine tailings as backfill of underground mine stopes results in an increase in their geotechnical stability, an increase in their geochemical stability, and a decrease in their leaching potential [25–28]. Comparatively, the conventional superficial management of mine tailings, stored within large storage dams together with high amounts

of water, has produced mine tailings that are in contact with air and precipitation, in some cases negatively affecting the quality of water in watersheds, due to acid rock drainage present in water seepage [29].

The environmental impacts of the surface disposal of mine tailings are increased if, when the mine tailings come into contact with the weather, acid rock drainage (ARD) and metal leaching are produced [30]. Due to this problem, many mining operations in Peru have chosen to store part of their mine tailings inside their underground mines.

The disposal of mine tailings in underground mine stopes (UMB) can limit the leaching of metals by reducing mine tailings porosity and decreasing their permeability, as well as providing an alkaline environment with the use of appropriate tailings neutralizing composition that can limit the potential for acid rock drainage (ARD) and metal leaching [31]. It is also in this way that another benefit, such as the elimination of particulate material to the surface environment, is achieved, reduction of the impact footprint of land use on the surface by not having a dam and reservoir such as tailings storage facilities and the elimination of seepage that can reach aquifers in watersheds [16,32].

The application of mine tailings as underground mine backfill (UMB) has been used globally in numerous mining operations [33–44]. Here are some examples:

- BHP Cannington, an underground silver, lead, and zinc mine in Northwest Australia, has operated a cemented paste backfill system since 1997.
- Stratoni Operations, at its Madem Lakkos and Mavres Petres lead, zinc, and silver mines in Greece has used cemented paste backfill since the 1990s.
- Higginsville Gold Mine, in Western Australia, has operated a cemented paste backfill plant since 2009.
- Barrick—Porgera, a gold mine in Papua New Guinea, uses cemented paste backfill in approximately 10% of the tailings from its operation.
- BHP Olympic Dam, a uranium and copper mine located in Australia, has used cemented paste backfill to manage a portion of its tailings and dumps.
- Barrick—Goldstrike, a gold mine located in Nevada, USA, has been using cemented paste backfill since 2013.
- El Toqui mine, located in Aysén, Chile, an underground gold and zinc mine, operates an integrated paste tailings management system, through which a part is used as cemented paste backfill for backfilling the underground mine.

### 1.2. Aim of the Article

A promising and increasingly popular management strategy for mine tailings in Peru is their disposal inside underground mines. This article presents technical characteristics and lessons learned from the practical experiences of tailings disposal in underground mines in Peru. In addition, some techniques are highlighted, such as (i) hydraulic fills, (ii) cemented hydraulic fills, and (iii) cemented paste backfill. Advantages and disadvantages are presented, together with a discussion of the scope, precautions, and application limitations of this solution. Finally, the main attributes of a green mining solution are presented, which allows for the reduction of socio-environmental impacts and improving the sustainability of stakeholders in the mining area, mainly due to: (i) mitigation of acid rock drainage (ARD), (ii) elimination of the risk of failure of tailings containment dams in the face of seismic hazard, (iii) reduction of potential seepage into watershed aquifers, (iv) elimination of particulate material emissions, (v) reduction of visual impact in front of large dams of containment, (vi) attenuation of the degradation of terrestrial habitats, and (vii) promotion of alternative uses of soils on the surface.

## 2. Different Types of Mine Backfilling Methods Using Mine Tailings Applied in Underground Mines from Peru

### 2.1. Characteristics of Backfilling Technologies in Peruvian Underground Mine Conditions

Peru is one of the countries with the largest number of mineral reserves, with mining deposits of copper, gold, silver, lead, tin, iron, and polymetallic resources. These deposits

are exploited through open-pit surface mining and underground mining. In general, mining deposits that have high mineral treatment productions above 50,000 mtpd are exploited by surface mining, while deposits with mineral treatment productions below 50,000 mtpd are exploited by underground mining.

Mine tailings produced by metallurgical processes must be stored in a safe and controlled manner. This is a complex and challenging task in Peru due to the complex topographic conditions present in the Andes, extreme weather conditions, being in a seismically active zone, and the existence of socio-environmental conflicts with the communities in the territories [29]. This has promoted the generation of alternative mining tailings management solutions to conventional depositing on the surface with the construction of large containment dams [12]. In this sense, the solution of storing all or part of the mine tailings inside underground mines has been positioned as an attractive alternative. This is how most of the underground mines in Peru use this solution and thus develop their mineral beneficiation tasks continuously and in harmony with the stakeholders in the territory.

The main function of backfill materials in underground mines is to help manage void-related mining stability, to increase the flexibility of ore extraction strategies, and often to allow for greater mineral recovery from mineral deposits [45]. The use of different types of backfill, their specific functions, and engineering requirements are closely related to exploitation methods, planning, and mining sequences [46,47].

Backfill material is used in underground mining operations for at least one of the following reasons:

- Recovery of rock pillars.
- Recovery of rock bridges.
- Work platform or floor.
- Support of the rock mass.
- Elimination of mine waste.
- To permit mining on top, to the side, or under fill.

The experience developed in Peru considering mine tailings as backfill for underground mines indicates that an adequate mining plan is important to coordinate mineral exploitation tasks with backfill activities. This is key to executing both activities without interference and continuously, always giving the operation of the underground mine a permanent functionality without delays or stoppages of the tasks.

Another aspect to consider for the entry of the mining tailings to be considered as an underground mine fill is the hydraulic transport in pipes, which can be carried out in some cases by gravity or by pumping [48]. This aspect will depend on the geometric configuration of the underground mine levels, taking into account whether the elevations in the masl of the underground mine exploitation levels will be below or above the elevation of the mine tailings backfill production plant for the underground mine [49].

Figure 2 shows a conceptual diagram of the operation of a Peruvian underground mine that considers a mine tailings backfill plant considering adit access, access ramps, transport galleries, different levels of mineral exploitation works, and the filling of stopes with mine tailings. In addition, in Figure 2, it is possible to appreciate the installations of bypass ramps, boreholes, ventilation shafts, and stopes or mineral exploitation areas to be filled with mine tailings material.

The entrance of the mine tailings to the underground mine is through pipes and must be made compatible with a series of activities that involve: (i) hauling and transporting ore with scoop machinery and mining trucks, (ii) energy conduction facilities with electricity for lighting services inside the underground mine, (iii) ventilation systems, (iv) compressed air intake facilities, (v) groundwater drainage pipes, (vi) fresh water intake pipes and (vii) fiber optic communications systems. Figure 3 shows a typical cross section of an underground mine horizontal tunnel implemented in mining operations in Peru:

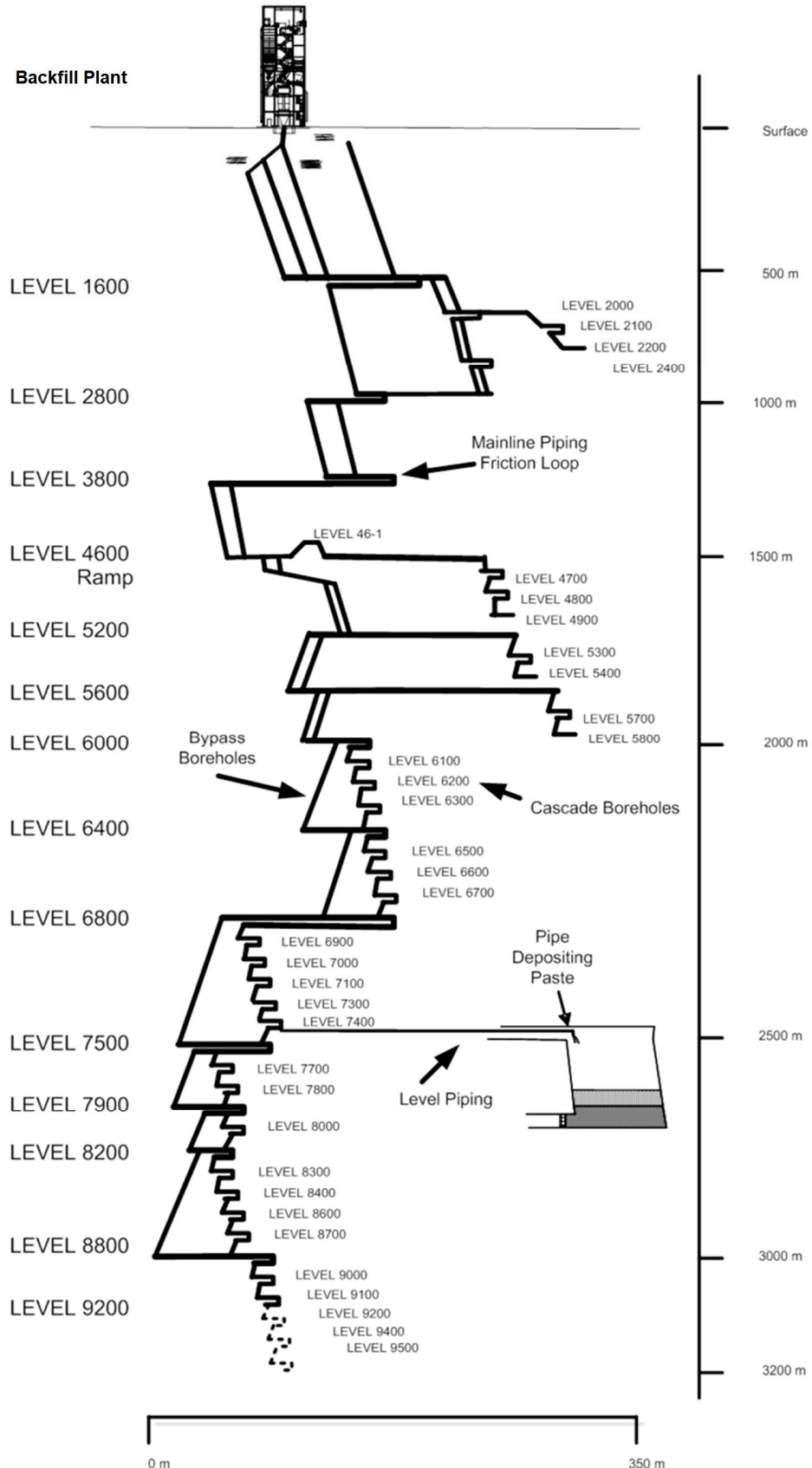

**Figure 2.** Typical schematic view of a Peruvian underground mine configuration using backfilling with mine tailings.

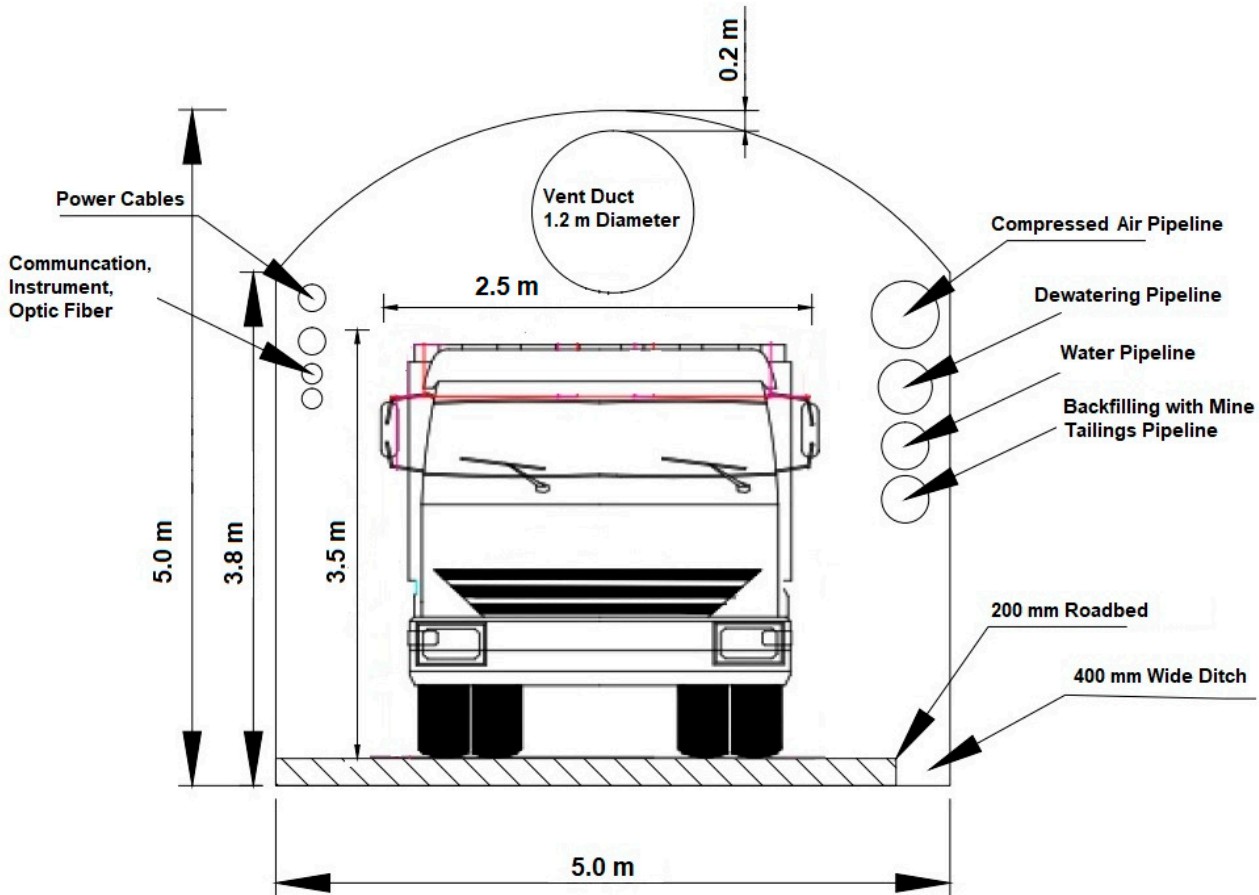

**Figure 3.** Typical cross section of an underground mine horizontal tunnel in Peru considering different services.

The consistent use of a properly engineered backfill system in an underground mine can allow higher extraction rates in each type of mineral deposit compared to an underground mine that does not use backfill. The improved safety conditions and the high extraction ratio are the direct benefits of the economic returns of the underground mine [50,51].

The mineral exploitation methods most used in underground mines in Peru are: (i) Sub Level Stoping, (ii) Cut and Fill, and (iii) Bench and Fill. These methods are executed in conjunction with the filling with mine tailings in order to exploit the underground mining deposits as much as possible.

Figure 4 shows an example schematic of a typical underground mine exploitation method executed in underground mines in Peru, where drilling, blasting, and haulage activities take place. Figure 5 shows a diagram of the configuration of access galleries to the different levels of the underground mine, where through piping systems, it is possible to enter the mine tailings fill for the different stopes. The stope fill is usually done from the top where the tailings are unloaded and thus proceed with the stope filling. Finally, Figure 6 shows a scheme of a stope filled with mine tailings at all its height considering different levels.

The filling of underground mine stopes with mine tailings requires an adequate design and rigorous operation [52]. The filling must occur under controlled safety conditions to avoid failures or accidents inside the underground mine [53]. Some of the aspects that must be considered are the following: (i) Geometry of the stope, (ii) Heat generation, (iii) Consolidation by own weight, (iv) Interaction of chemical components of the mine tailings, (v) Rheology of mine tailings, (vi) Interfacial interaction, (vii) Stability conditions of the stope walls, (viii) Stress distribution, (ix) Behavior of dynamic loads, (x) Loads in plugs or barricades, in accordance with what is indicated in Figure 7.

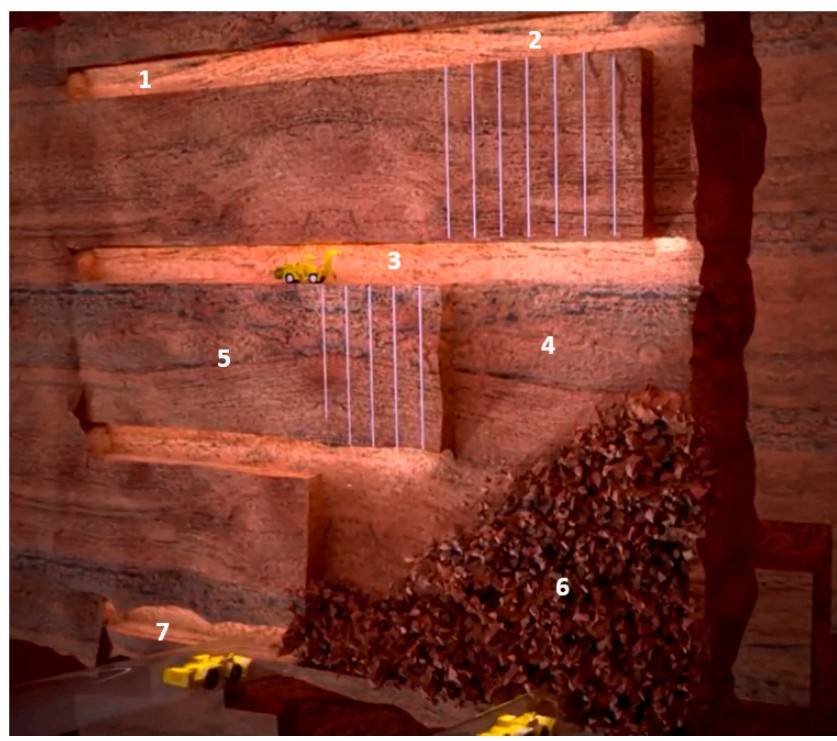

**Figure 4.** Typical underground mine exploitation method applied in Peru. (**1**) Sublevel drift, (**2**) Drilling, (**3**) Drilling and blasting, (**4**) Stope, (**5**) Solid ore, (**6**) Blasted ore, (**7**) Loading crosscut.

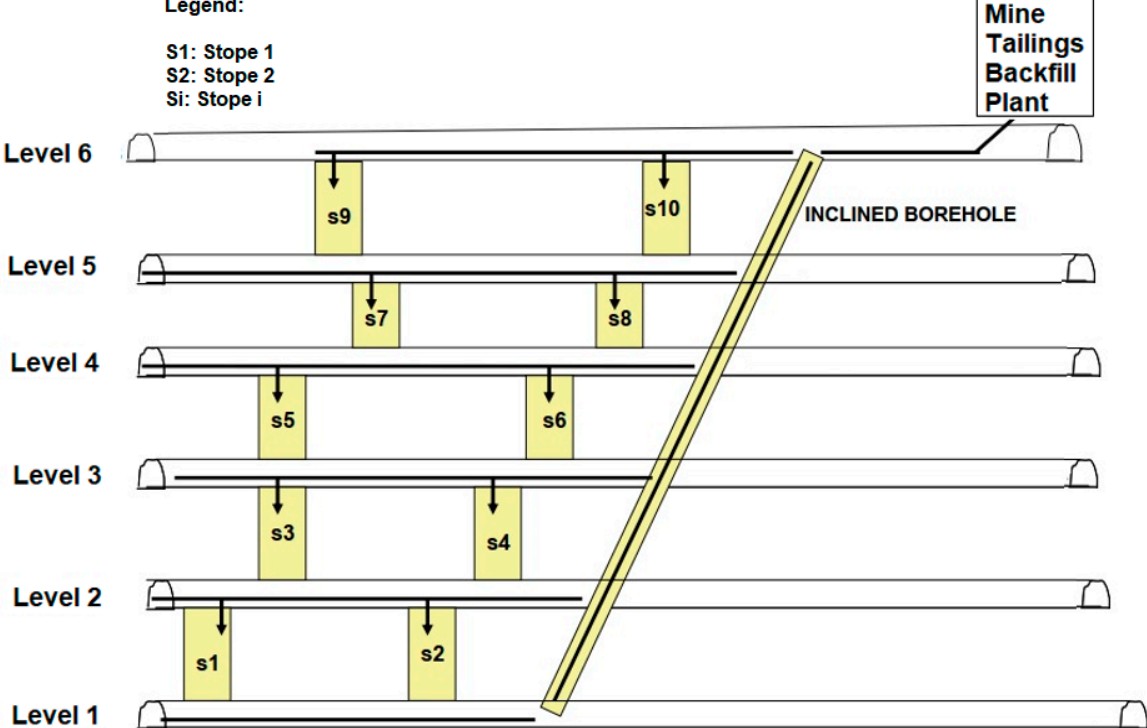

**Figure 5.** Typical configuration of mine tailings backfill distribution on stopes in an underground mine in Peru.

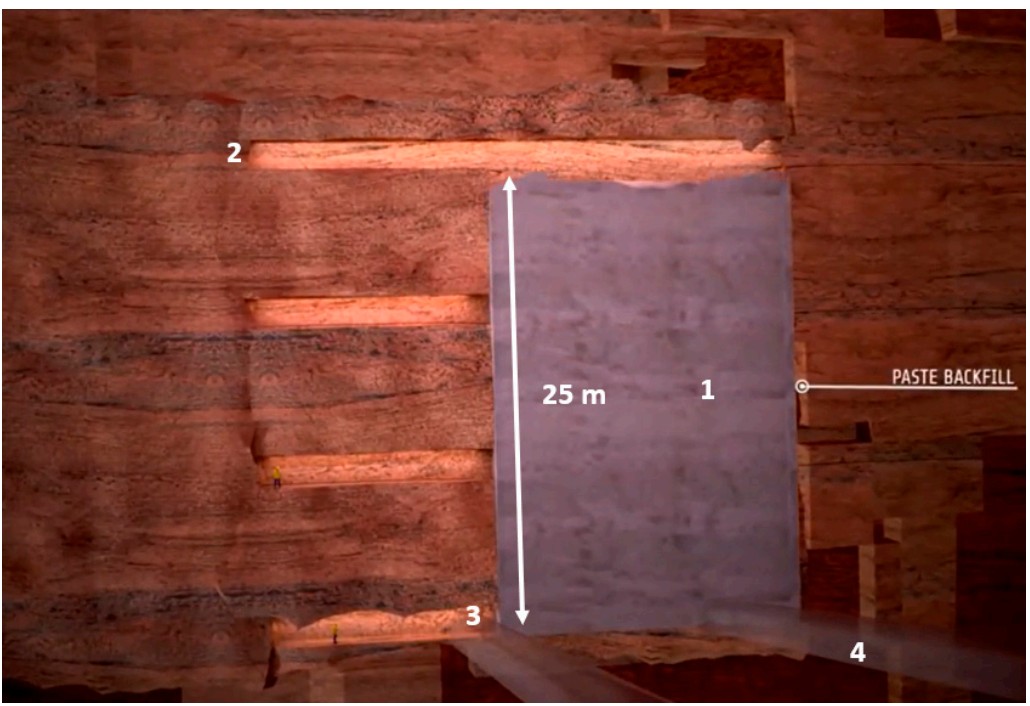

**Figure 6.** Typical view of stope filled with paste backfill. (**1**) Filled stope, (**2**) Sublevel drift, (**3**) Rockpile barricade, (**4**) Loading crosscut.

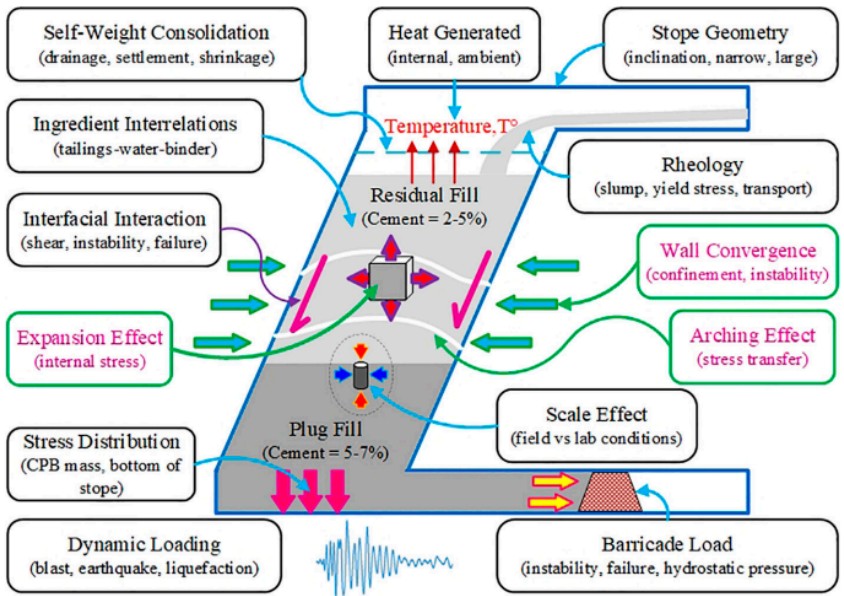

**Figure 7.** A schematic view of backfilling of stope with mine tailings and aspects to be considered [52].

Underground mine fills can be grouped into two general categories: non-cemented, which includes hydraulic and mineral paste fills, and cemented mine fills, which include the addition of binding agents such as Portland cement or Portland cement mixtures along with other pozzolanic additions, fly ash, gypsum, or slag [54,55].

The most common forms of underground mine fill with mine tailings include (i) hydraulic fill, (ii) cemented hydraulic fill, and (iii) cemented paste backfill. Each type of backfill has its own risks that need to be addressed during underground mine design, planning, and operations [56]. Of all the threats posed by backfill to underground mining, the biggest threat depends on the water content in the backfill [31]. Figure 8 shows an example of hydraulic underground mine backfilling with mine tailings applied in Peru.

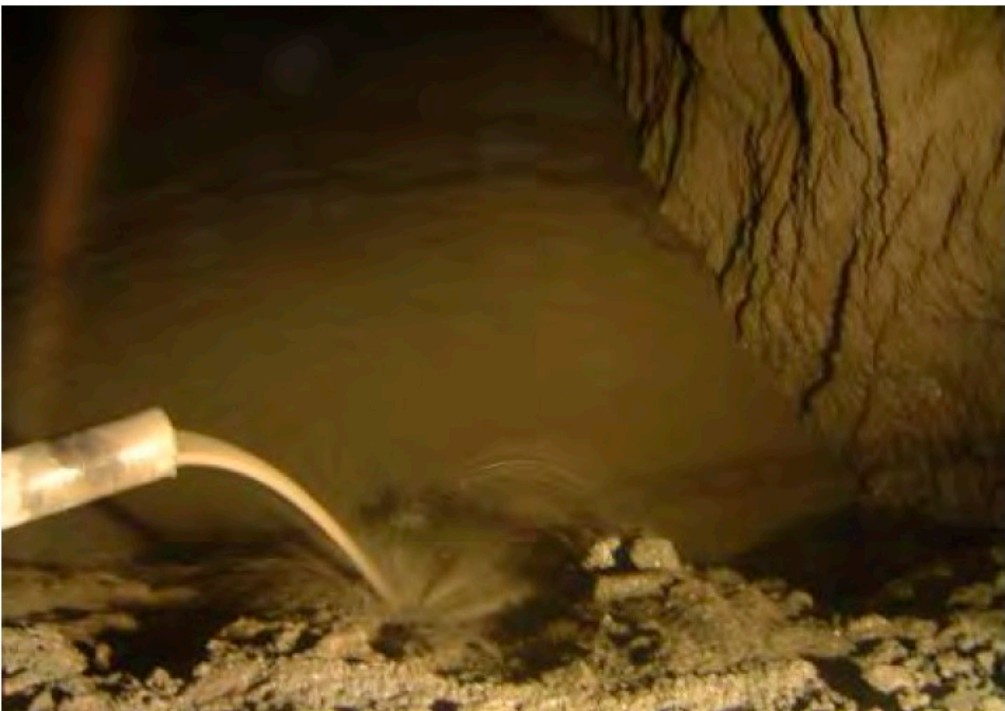

**Figure 8.** Example of backfilling with mine tailings of stope in an underground mine in Peru.

## 2.2. Hydraulic Fill

Hydraulic fill makes use of the coarse fraction of mine tailings obtained through a granulometric classification process using hydrocyclones (cycloned tailings sand). It is an alternative to filling in the pits of the underground mine exploited by the Cut and Fill method; this alternative, used in Peru since 1937, was introduced at the beginning with the aim of increasing productivity, but not for socio-environmental purposes. The hydraulic fill replaced the detrital fill or solid fill transported in mining cars or trucks from the quarries to the stopes and at a much higher cost than the other filling methods that are carried out by transporting the solids of mine tailings, for example, with hydraulic transport by pipeline and driven by pumps.

Hydraulic fill refers to the fillings that are transported as a high-density slurry through pipes to underground works. The hydraulic fill is prepared with the cycloned and deslimed mine tailings, where the maximum particle size is 1 mm, and the content of fine particles smaller than 10 μm should not be greater than 10% of the total mass of the mine tailings. This hydraulic fill requires a concentration of solids by maximum weight Cw equal to 70%. Like the dry fill, it is relatively cheap; however, it is necessary to be careful when depositing it since there may be safety problems due to its high-water content with respect to permeability, drainage, liquefaction, and to the danger of pipeline blockages [57].

The hydraulic fill, as with any method of filling, has two main purposes in the underground mining operations carried out in Peru: (i) the first is to serve as a working floor to carry out the drilling, blasting, and hauling of ore operations, and (ii) the second is to serve as support so that the underground mine does not collapse due to the increase in open areas. Figure 9 shows a typical flow diagram of the underground mine hydraulic fill production and transportation process applied in Peru:

The preparation for the entry of the fill and the transport of the fill are part of the underground mining stages within the exploitation cycle by the Cut and Fill method, either ascending or descending. These activities of preparation for filling and transporting the filling occupy 30% to 40% of the time spent within the cycle. Due to the speed with which the activities of the underground mining cycle must be carried out, the fill must meet certain granulometry and percolation speed or permeability index requirements. Generally, the tailings generated by the concentrator plant are used, which are conducted to a particle

size distribution classification stage through the use of hydrocyclones in order to obtain cycloned mine tailings (with particle sizes greater than ASTM # 200 mesh). which are stored in a tank with water where it is mixed by means of an agitator. This mixture of cycloned sands from the mine tailings with water is the hydraulic fill, which is pumped, in order to be transported through a double-layer high-pressure pipeline, with the inner wall of high-alloy steel for the purpose of lining and reduction of pipe thickness due to accelerated wear. The power of the pump and the diameter of the pipe are calculated based on the conditions and requirements that are particularly present in each mine.

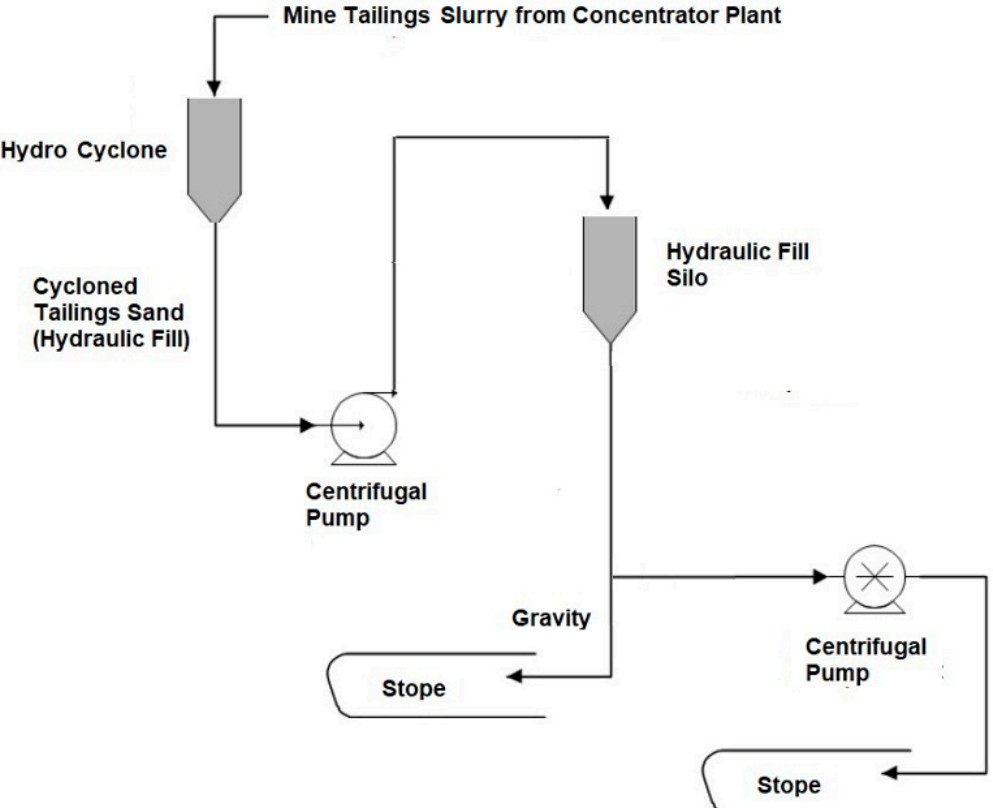

**Figure 9.** Typical process diagram flow of production and transport of hydraulic fill applied in underground mines in Peru.

The fine tailings called slimes, resulting from the use of hydrocyclones, are sent to a surface mine tailings storage facility through the construction of a reservoir and a dam. These materials are not used as hydraulic fill because they retain a lot of water due to their low permeability, and also, due to the presence of very fine particles, they do not provide appropriate geotechnical resistance as fill in underground mines.

The potential underground deposits can be all the stopes finished or emptied of the fractured ore, as long as they have been previously planned and prepared for this purpose. Open-stope mining methods are ideal for depositing cycloned tailings sands. Deposits exploited by the Cut and Fill method could also be redesigned in order to extract the ore by vertical slices, either exploiting the ore with controlled blasting, with the purpose of extracting the ore without damaging the surrounding rocks, leaving the chambers open and suitable to deposit the cycloned tailings sands. The stopes would always be filled so that this stage is completely detached from the mining cycle, and the cycloned tailings sands would be sent to the underground mine, just as it is produced by the concentrator plant. If the stopes to be filled were at the same level or above the level of the flotation cells of the concentrator plant, experience has shown that it has always been necessary to pump the cycloned tailings sands to underground deposits through a network of pipes; Otherwise, the cycloned tailings sands would travel by gravity in pipelines (Figure 10).

Each case of exploitation will require a particular design to prepare the finished stopes as storage places for the cycloned tailings sands.

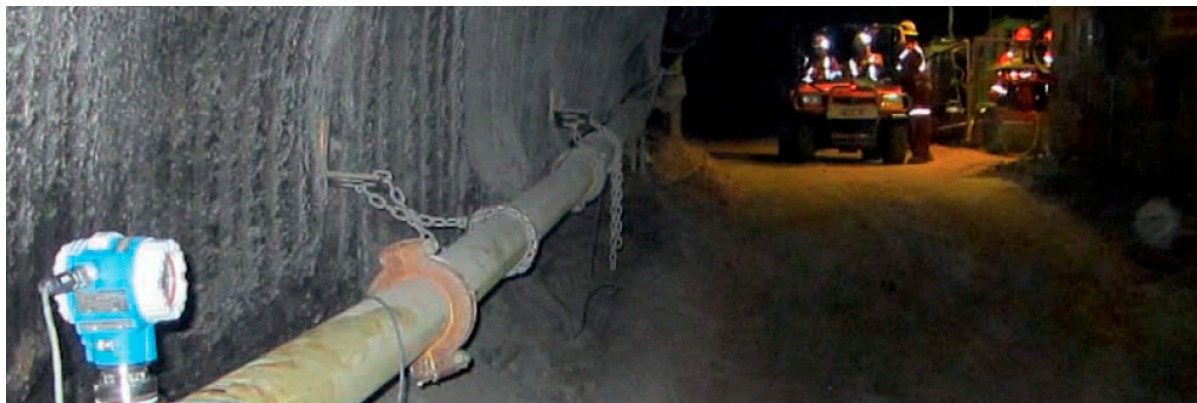

**Figure 10.** Example of transport of hydraulic fill in an underground mine in Peru.

*2.3. Cemented Hydraulic Fill*

Properly prepared uncemented hydraulic fill is predominantly a graded aggregate product and should behave like sand, although it may occasionally behave like slow-draining sand. Consequently, it has no true cohesion until drained, nor does it have unconfined compressive strength; it is generally desirable that the filler be able to remain in an area after exposure. For such applications, it is customary to add cement to the fill.

Because cemented hydraulic fill is the same material as uncemented, with the exception of cement, many of the design parameters for a cemented hydraulic fill system are the same as those for an uncemented fill system. However, cement provides fines to the mix, and the rate of drainage is typically somewhat slower than that of uncemented fill.

The addition of cement allows improvements in the compressive strength of the filling, which is important when the filled cavity is used as a pillar or base of an exploitation level. The cost advantage of the uncemented method is lost by using cement. Figure 11 shows a typical flow chart of the underground mine cemented hydraulic fill production and transportation process applied in Peru:

The cemented hydraulic fill operation moves from the underground disposal of mining waste to the preparation of an engineering material controlled for the quality of cement content versus strength and deformation properties.

The relevant properties and parameters of the cemented hydraulic fill are as follows:

- Ratio of voids and porosity.
- Relative density.
- Permeability.
- Cut resistance.
- Effective efforts.
- Apparent, saturated, and submerged unit weight.
- Lateral earth pressure.
- Filtration, drainage.
- Liquefaction.
- Slurry rheology.

The transport of cemented hydraulic fill is mainly in the form of slurry through HDPE or Steel pipes. The cemented hydraulic fill slurry is transported into the underground mine using the force of gravity or external energy generated by pumping systems. The cemented hydraulic fill does not require the additional use of machinery for its distribution in the stopes of the underground mine due to the advantage of the slurry that seeks its own level of fill and fully occupies all the spaces and fractures generated by blasting in the stopes (Figure 12).

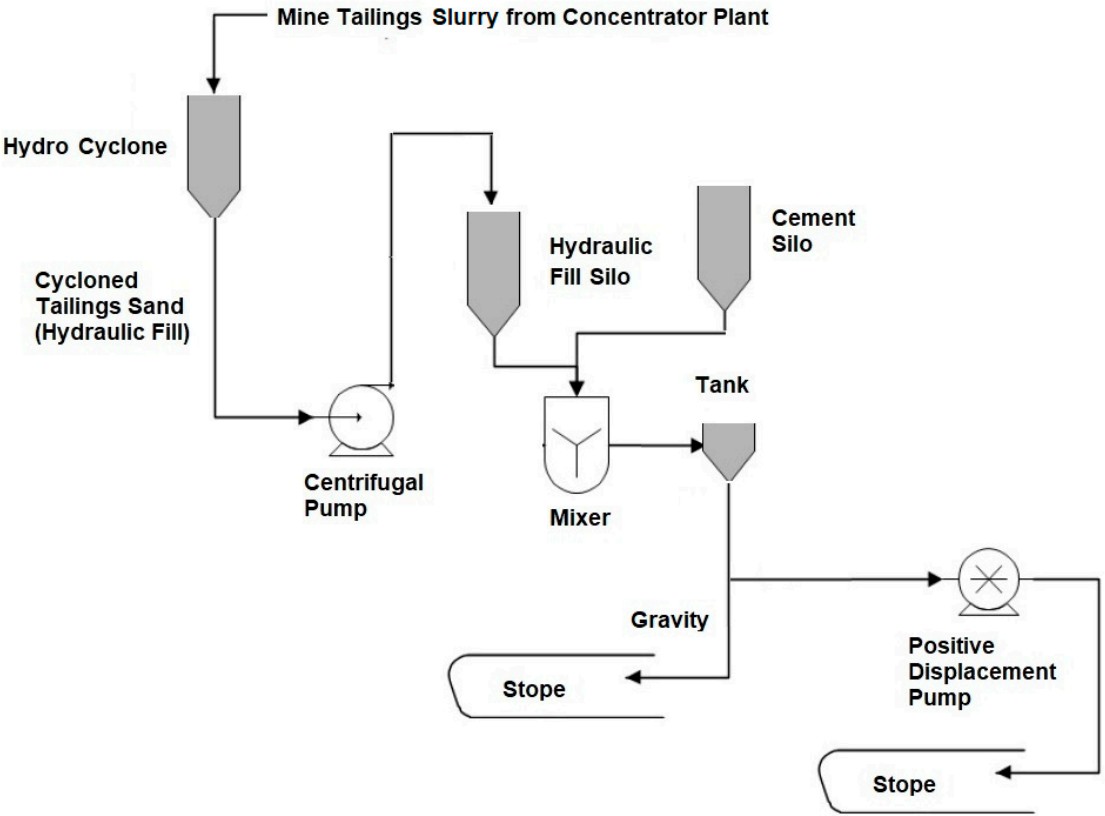

**Figure 11.** Typical process diagram flow of production and transport of cemented hydraulic fill applied in underground mines in Peru.

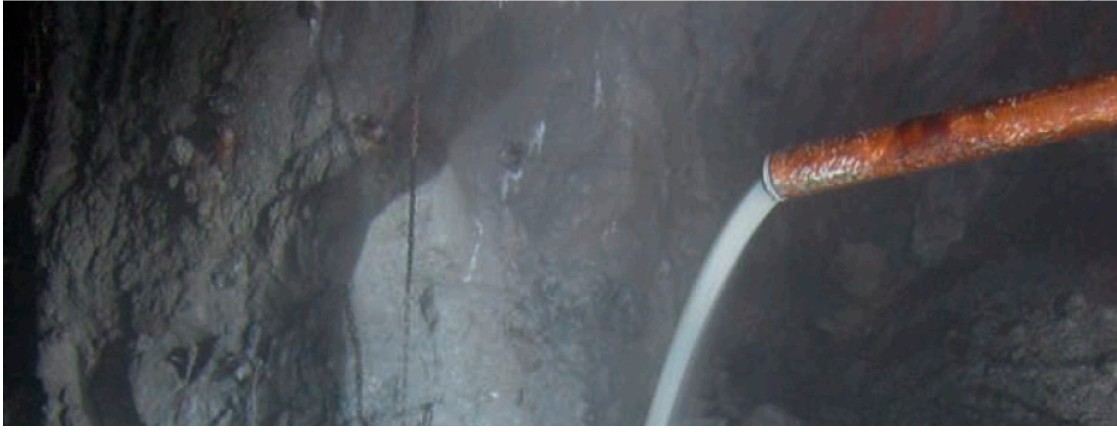

**Figure 12.** Example of disposal of cemented hydraulic fill in an underground mine in Peru.

The main applications of cemented hydraulic fill are: (i) to provide a strong working platform inside the underground mine, (ii) to stabilize the rock mass and thus reduce the probability of the presence of the phenomenon known as rockburst, (iii) to facilitate the recovery of ore from pillars and safety bridges, and (iv) to prevent movements, landslides, or rock falls from level to level (subsidence). The cemented hydraulic fill technique suits underground mine sub-level stoping and chamber-and-pillar mining methods.

The cemented hydraulic fill is a mixture of tailings plus water plus cement, the latter being in an approximate dosage within the range of 50–100 Kg per m$^3$. The cemented hydraulic fill has consistency and characteristics like the normal hydraulic fill, but the difference is that the necessary mechanical resistance (0.5 MPa) is achieved in less time than the hydraulic fill (21 days).

*2.4. Cemented Paste Backfill*

In this chapter, the application of cemented paste tailings technology for the filling of underground mines will be deepened, considering that it is an application of cement with attractive development potential in world mining and in Peru. Likewise, the surface disposal of paste tailings currently has important advances worldwide and in applications in medium-sized mining and, incipiently, in large-scale mining in Peru.

Cemented paste backfills are composed of mine tailings plus cement plus water plus additives. The cemented paste backfill usually has the following design proportions: mine tailings 1700 Kg + cement 95 Kg + water 500 L + additive 1.5 L and water/cement ratio 5.0. In particular, mine tailings used for backfilling underground mines such as a cemented paste backfill must first be thickened to form mineral paste and then cemented by adding small amounts of cement (in the order of 3 to 8% by weight) in such a way that adequate mechanical resistance is achieved. There must be a sufficient amount of fines (<20 μm) in the mine tailings for it to have paste behavior. Its properties are the same as a cemented hydraulic fill, but in this case, the paste rheology must be considered; that is, the behavior of the paste itself is incorporated for the analysis of its transport. Subsequently, the paste must necessarily flow through a pipeline with the help of the force of pumping systems, to later be arranged horizontally when carrying out the filling operation of the underground mine. The nature of paste backfill products (i.e., high viscosity) precludes the use of gravity distribution capabilities that are related to the use of hydraulic backfills; without the use of positive displacement pumps at the surface manufacturing sites, the transport of paste fills becomes highly problematic. Gravity feed of paste fills is not viable in most cases unless excessive water contents are used in the paste tailings material, thereby relegating it to a high-density tailings product and not what is commonly considered to be a paste material. Figure 13 shows a typical flow diagram of the production and transportation process of cemented paste fill in an underground mine applied in Peru:

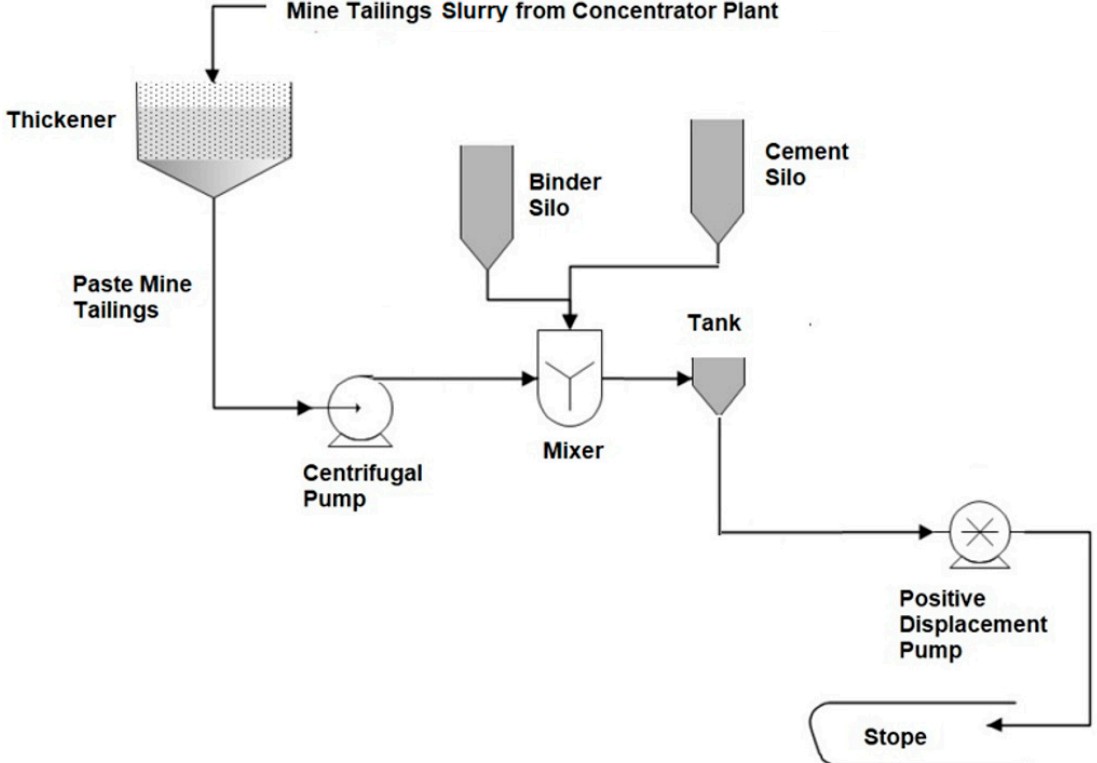

**Figure 13.** Typical process diagram flow of production and transport of cemented paste backfill applied in underground mines in Peru.

With the cemented paste backfill filling the galleries of the underground mines, it is possible to increase their local stability, which brings, as a consequence, an increase in the safety and efficiency of mining operations. For example, filling with cemented paste backfill allows the extraction of mineralized pillars when the so-called "tunnels and pillars" method is used in mining. This also prevents the occurrence of landslides inside the mine. The inclusion of cemented paste backfill has the additional advantage of reducing or eliminating the impact on the surface associated with the disposal of tailings stored with large dams. Figure 14 shows the fill of a stope with cemented paste backfill in an underground mine in Peru.

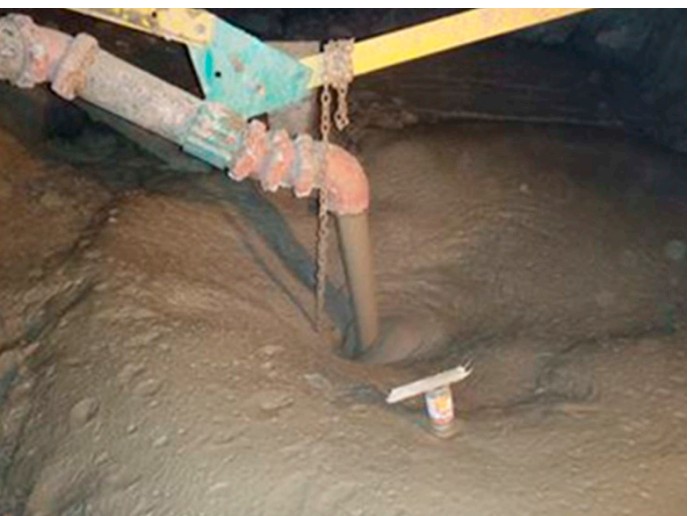

**Figure 14.** Example of disposal of cemented paste backfill in an underground mine in Peru.

Considering the requirement and demand for filling an underground mine with cemented paste backfill, the characteristics of the production plant can be of 2 types: (i) operation in Batch mode or (ii) operation in continuous mode [58]. Figures 15 and 16 show examples of typical flowsheets for underground mine backfill plants with cemented paste backfill applied in Peru considering Batch operation and continuous operation, respectively.

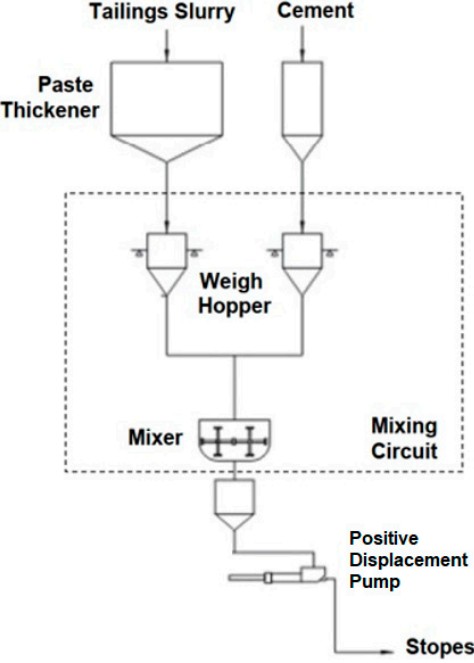

**Figure 15.** Example of Batch process flow diagram of cemented paste backfill plant applied in Peru.

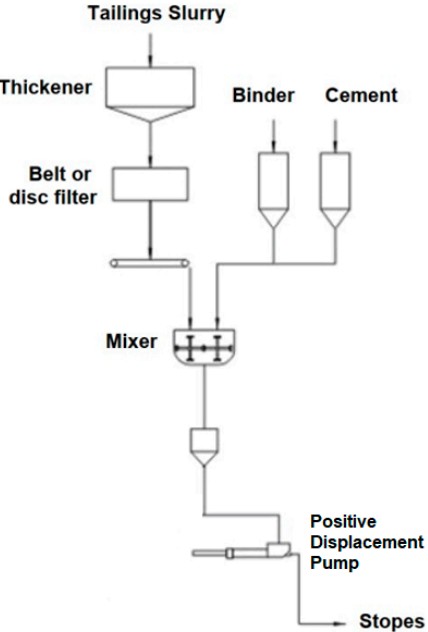

**Figure 16.** Example of continuous process flow diagram of cemented paste backfill plant applied in Peru.

According to the consistency and rheology of non-Newtonian behavior of the underground mine fill material made up of cemented paste backfill, a material with complex hydraulic behavior can be obtained to be transported through pipelines [59–61]. Due to the high viscosity, the high yield stress of the material being of the order of at least 125 Pa, and demanding pumping pressure requirements of the order of 125 Bar (12.5 MPa) in some cases, the use of positive displacement pumps is required (PD Pumps) [12]. These conditions do not make possible the use of centrifugal pumps, making the hydraulic piston type of positive displacement pump the most widely used in the underground mining industry in Peru. Positive displacement pumps make it possible to ensure a pulsed transport of the material through the pipe network in a laminar hydraulic regime, avoiding clogging of material and accelerated wear of the pipe wall thickness.

An easy and popular method used in Peru by operators of underground mine fill plants using cemented paste backfill to indicate flow behavior is the standard 12-inch (305 mm high) slump cone developed for measurements of concrete settlement (ASTM C143/C 143M-00). This test can be used to obtain a preliminary evaluation of the flow behavior of the cemented paste tailings pipeline and allows quality control for the transport of the material to the stopes. Experience with pilot and large-scale operations suggests that for slumps greater than 9 inches (230 mm), the lower yield strength of cemented paste backfill will be a pumpable fluid. For slumps between 8 and 9 inches (200–230 mm), the yield strength will be high (Ty > 150 Pa), and the possibility of a shear system should be evaluated. For slumps of 8 inches (200 mm) and less, shearing will be required (Ty > 300 Pa). These operating ranges are strongly determined by solid particle size, throughput rate, and pumping/piping shear [62–64].

### 3. Selection Criteria and Design Issues for Underground Mine Backfill Systems Using Mine Tailings Considering Experiences Developed in Peru

Considering the practical experience acquired in Peru in the management of mine tailings as backfill for underground mines over the last 50 years in different large-scale mining operations, Table 1 is presented, showing suggested selection criteria and a comparative analysis with the main attributes of the three main techniques implemented in the Andean mining zone of Peru that considers: (i) hydraulic fill, (ii) cemented hydraulic fill, and (iii) cemented paste backfill.

**Table 1.** Selection criteria, design issues, and comparative analysis of different types of backfilling methods applied in Peru.

| Characteristics/Type of Fill | Hydraulic Fill | Cemented Hydraulic Fill | Cemented Paste Backfill |
|---|---|---|---|
| Mix of the slurry | Cycloned Tailings Sand + Water | Cycloned Tailings Sand + Cement + Water | Mine Tailings + Cement + Binder + Water |
| Concentration of solid by weight of the slurry (Cw) | 60–75% | 70–80% | 75–85% |
| Use of cement | No | Yes (according to mix design and geomechanics recommendation) | Yes (according to mix design and geomechanics recommendation) |
| Type of mine tailings | Classified mine tailings (cycloned tailings sands) | Classified mine tailings (cycloned tailings sands) + quarry material | Paste mine tailings + slags |
| Type of transport | HDPE pipeline | HDPE pipeline/Steel pipeline | Steel pipeline |
| Pumping equipment | By gravity/Centrifugal pumps | By gravity/Centrifugal pumps/Positive displacement pumps | Positive displacement pumps |
| Compression strength (Geomechanics) | Low < 0.3 MPa | According to the mix design: 0.6 < Strength < 4.0 MPa | According to the mix design: 0.8 < Strength < 3.5 MPa |
| Underground mine methods applied | Ascending cut and fill | Chambers and pillars, Cut and fill descending, Sub level stoping | Sub level stoping, block caving |
| Plug and barricade system | Mine waste rock dyke with appropriate draining barricade systems | Mine waste rock dyke and concrete walls and appropriate drainage systems | Concrete walls |

Table 1 provides recommendations for the selection of backfilling technique, design of backfilling of the underground mine, and backfill transport systems using mine tailings considering conservative criteria based on the experience obtained in practice from the reality of mining in Peru. Some aspects to consider in the selection of the type of backfill and when evaluating the implementation of these engineering systems are the following:

- Mine tailings slurry mixture: It must be defined according to the process diagram of the mining operation.
- Percentage of solids in the mine tailings slurry: It must be defined according to the type of fill to be implemented, be it hydraulic fill, cemented hydraulic fill or cemented paste backfill.
- Use of cement: It must be evaluated according to the requirements of geomechanical resistance of the filling of the stopes inside the underground mine.
- Type of tailings to be used: It must be defined according to the type of fill to be implemented, be it hydraulic fill, cemented hydraulic fill, or cemented paste backfill.
- System of transportation: It must be defined considering the wear rate of the pipe material, which will mean an operational cost for changing pipes.
- Pumping equipment: It must be defined according to the discharge pressure requirements of the hydraulic system, taking into account the rheology of the material to be transported, changes in topographic level, and length of the pipes.
- Resistance to compression: It is defined considering the geomechanical requirements of the filling of the stopes.
- Underground mining methods: They are defined according to a technical-economic feasibility analysis of the optimal exploitation of the underground mine.
- Containment system or barricades: They are defined according to the static and dynamic load requirements of the mansion fill on the walls or barricades or caps to be implemented.

For the design of underground mine backfill engineering systems with the use of mine tailings, it is recommended to consider the following methodological procedure in stages in order to define key processes, size equipment, size piping systems, and verify the supply-demand of the backfill for underground mine. The main stages to consider in the design of these systems are shown in Figure 17, and these are mainly: (i) Physical and mechanical characterization of the backfill, (ii) Mining plan, (iii) General backfill criteria of stopes, (iv) Sizing and process design of the backfill plant and (v) Sizing and hydraulic design of the mine tailings backfill transportation system.

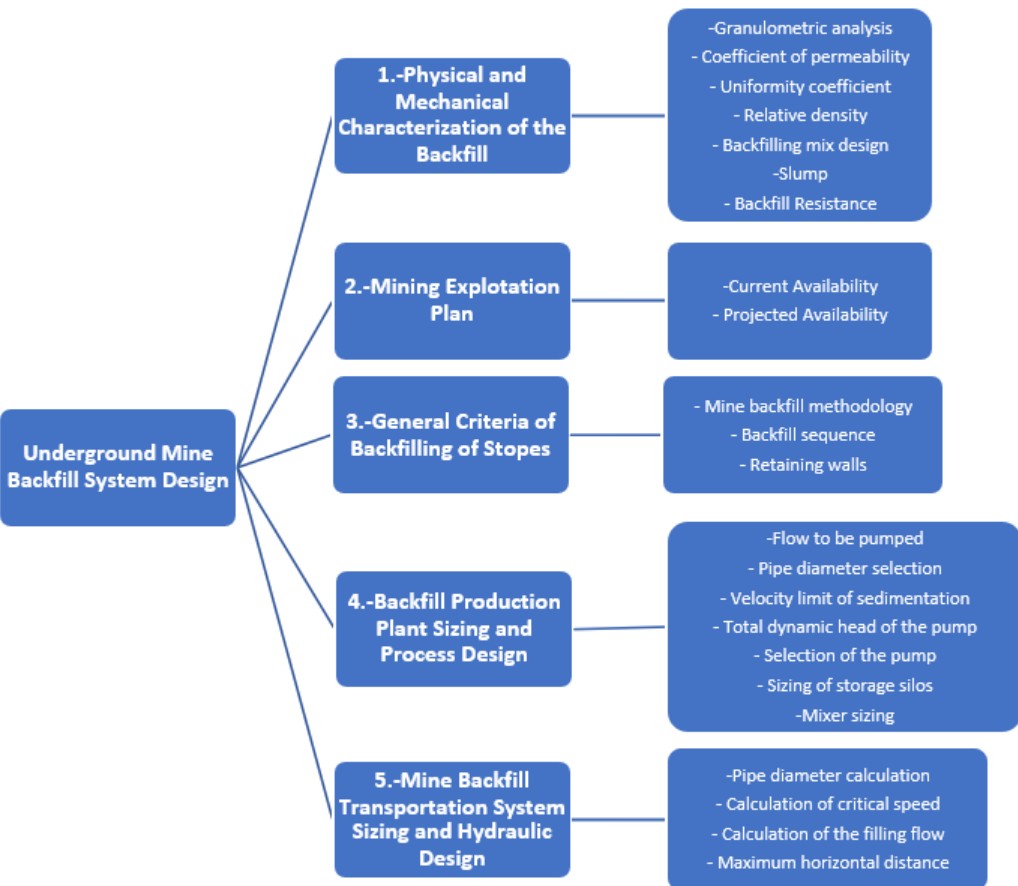

**Figure 17.** Methodological procedure for the design of backfilling with mine tailings in underground mine engineering systems.

Considering the main stages to consider in the design of these systems are shown in Figure 17, the following lessons have been learned:

- Physical and mechanical characterization of the backfill: This is a key stage in the design of the system since it will allow us to understand the behavior of the material throughout the process.
- Mining exploitation plan: It will define the useful life of the system to be designed and its form of implementation, either in different phases over time or in different sectors of the underground mine.
- General criteria for the filling of stopes: This aspect is important for the geomechanical conditions to take into account in stopes and to provide safety in underground mine operations.
- Sizing and process design of the backfill production plant: The proper sizing of the process plant is relevant since it has an impact on the capital cost of the project.
- Sizing and hydraulic design of the mine backfill transportation system: Its correct sizing is essential to avoid incurring high operating costs due to the constant change of pipes due to accelerated wear and/or breakage.

Considering the experience in engineering designs of underground mine backfill process plants with the use of mine tailings in Peru, Figure 18 presents a 3D model with the main facilities and electromechanical equipment required to produce cemented hydraulic fill.

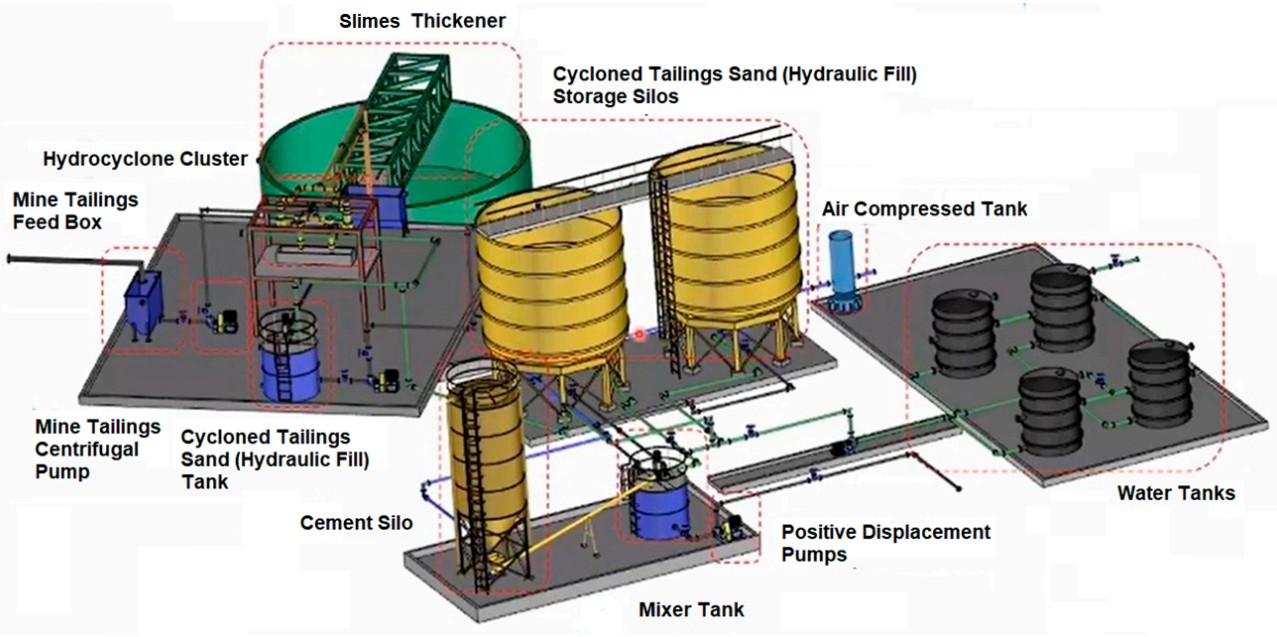

**Figure 18.** Example of 3D drawing of cemented hydraulic fill process plant designed in Peru.

Figure 18 shows a process where the mine tailings from the concentrator plant are classified through hydrocyclones, where the fine fraction obtained called slimes are conducted to a thickener to increase its density and recover water from reusing in the metallurgical process, where the thickened slimes are finally pumped to a surface tailings storage facility. On the other hand, the coarse fraction obtained by the hydrocyclones called cycloned tailings sands, is transported to a tank where it is homogenized and then stored in two silos. As hydraulic fill is required in the underground mine, the cycloned tailings sands are released from the silos and directed to a mixing tank, where they are mixed with cement. This mixture obtained between the cycloned tailings sands and the cement is called the cemented hydraulic fill. Finally, the mixture obtained is driven by a positive displacement pump (PD Pump) through pipes towards the stopes of the underground mine. Finally, Figure 19 shows a real image of the described process plant that has been implemented in Peru.

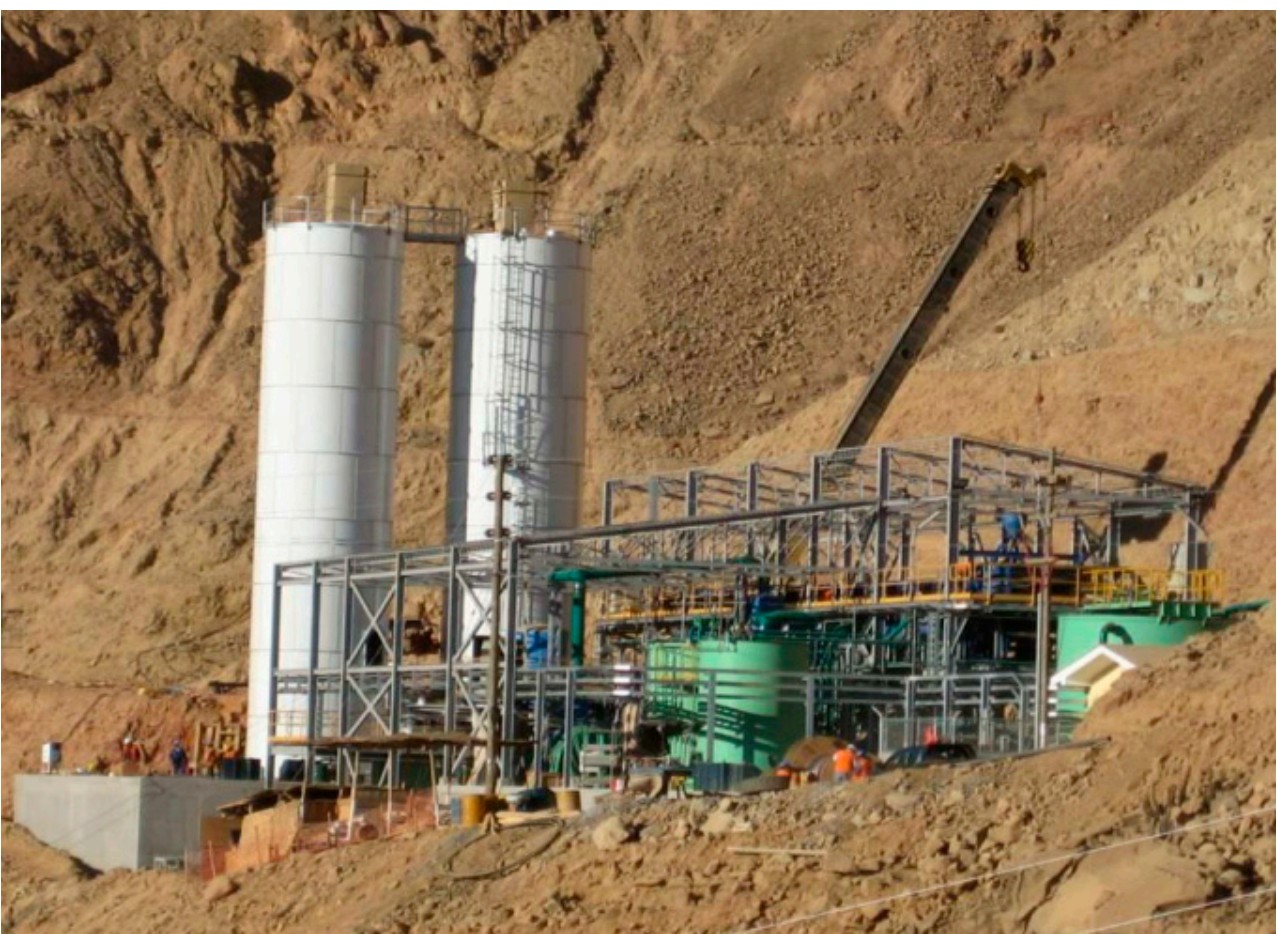

**Figure 19.** Panoramic view of cemented hydraulic fill process plant in an operation in Peru.

## 4. Cases Studies—Practical Experiences Developed in Peru

The main underground mining operations in Peru are located mainly in the Andean zone, in the middle of the Andes Mountain range with elevations between 4000 masl and 5000 masl. In this sense, the topographic and climatic conditions are complex due to the presence of terrain with valleys with steep mountain slopes and the presence of frost/rain/hail/snowstorms. In addition to these restrictions on the normal mining operation, there are demands from the communities for environmental conditions free of contamination, where past bad practices with cases of mining wastes generating acid rock drainage have created a bad reputation for the mining industry in Peru, which affects a continuous improvement of the current daily mining operational management. These conditions strongly restrict, in some cases, the depositing of mine tailings on the surface, promoting the use of mine tailings as a backfill for underground mines.

In Peru, there are several underground mining operations using mine tailings as backfills, in different Andean regions of Peru, for different mining deposits, mainly of the polymetallic type. A comparison of the main characteristics of the mining operations is presented in Table 2. This table includes (i) Mine name, (ii) Mining company, (iii) Beneficiation metals, (iv) Location, (v) Mine tailings production, (vi) Underground mine exploitation method, and (vii) Underground mine backfill method.

**Table 2.** Main characteristics of backfilling with mine tailings of underground mining activities in Peru. The following abbreviations are considered for the Underground Mine Exploitation Method (Ascending Cut and Fill, AC&F, Descending Cut and Fill, DC&F, Bench and Fill, B&F, or Sub Level Stoping, SLS), and Backfill Method (Hydraulic Fill, HF, Cemented Hydraulic Fill, CHF or Cemented Paste Backfill, CPB).

| Name of Mine | Mining Company | Metals | Location | Mine Tailings Production (mtpd) | Underground Mine Exploitation Method | Backfill Method |
|---|---|---|---|---|---|---|
| Cerro Lindo | Nexa | Cu, Zn, Pb | Ica | 10,000 | SLS | CPB |
| El Porvenir | Nexa | Cu, Zn, Pb | Pasco | 6000 | AC&F | HF |
| San Rafael | Minsur | Sn | Puno | 2000 | SLS | CPB |
| Andaychagua | Glencore | Cu, Zn, Pb | Junín | 2000 | DC&F | CHF |
| Paragsha | Glencore | Cu, Zn, Pb | Pasco | 8000 | DC&F | CHF |
| Carahuacra | Glencore | Cu, Zn, Pb | Junín | 3000 | B&F | HF |
| Animón | Glencore | Cu, Zn, Pb | Pasco | 2000 | AC&F | HF |
| San Cristóbal | Glencore | Cu, Zn, Pb | Junín | 1500 | AC&F | HF |
| Uchucchacua | Buenaventura | Ag, Pb, Zn | Lima | 1500 | AC&F | HF |
| Orcopampa | Buenaventura | Au, Ag | Arequipa | 1500 | AC&F | HF |
| Tambomayo | Buenaventura | Au, Ag | Arequipa | 1000 | B&F | HF |
| Cobriza | Doe Run | Cu, Zn, Pb | Huancavelica | 2500 | AC&F | HF |
| Inmaculada | Hochschild | Au, Ag | Ayacucho | 1500 | DC&F | CHF |
| Pallancata | Hochschild | Au, Ag | Ayacucho | 1500 | DC&F | CHF |
| Jimena | Poderosa | Au, Ag | La Libertad | 1000 | AC&F | HF |
| Parcoy | Horizonte | Au, Ag | La Libertad | 1500 | AC&F | HF |

The information provided by Table 2 reveals the following results:

- The production of mine tailings considered for backfilling underground mines is in the range of 1000 to 10,000 mtpd, considering small and medium-scale mining operations.
- The bench and fill method of underground mining is compatible with the use of hydraulic fill.
- The ascending cut and fill method of exploitation of underground mining is compatible with the use of hydraulic fill.
- The descending cut and fill method of underground mining is compatible with the use of cemented hydraulic fill.
- The sub-level stoping underground mining exploitation method is compatible with the use of cemented paste backfill.
- The most popular underground mine backfill method applied in Peru is hydraulic fill, followed by cemented hydraulic fill, and finally, cemented paste backfill.

## 5. Discussion

The filling of underground mines with mine tailings from the stopes helps in: (i) the structural support of the underground mine, (ii) Reduces the risk of explosion or rock burst because the pressures in the rock are not concentrated in the pillars and supports of the mine, (iii) Improves the ventilation circuit in the mine, (iv) Prevents falls from the mine roof before blasting, (v) The binder helps to minimize the contamination of groundwater and (vi) The development of acid rock drainage (ARD) is reduced.

Some of the most important applications of mine tailings backfill in underground mines are the following:

- Work platform.
- Avoid landslides and falling rocks.
- Facilitate the recovery of pillars.

- Avoid or minimize subsidence.
- Stabilize the rock mass in underground mines.
- Reduce the possibility of rock bursts.
- Minimize the use of wood.

The main advantages and disadvantages of the use of hydraulic fill, the use of cemented hydraulic fill, and the use of cemented paste backfill are presented in Table 3 below:

**Table 3.** Advantages and disadvantages of hydraulic fill, cemented hydraulic fill, and cemented paste backfill application.

| Hydraulic Fill | |
| --- | --- |
| **Advantages** | **Disadvantages** |
| - Cost-effective method. | - Drainage and water management system inside underground mine is required. |
| - Easy application method. | - Flood of underground mine excavations are possible. |
| - In some cases, pumping is not required. | - Pipeline plugging. |

| Cemented Hydraulic Fill | |
| --- | --- |
| **Advantages** | **Disadvantages** |
| - Pipeline transport is much more economical, efficient, and faster than with other types of transport. | - The cemented hydraulic fill system requires a higher capital investment than hydraulic fill. |
| - When the backfill is deposited in the stope in the form of slurry, it seeks its level naturally, thus eliminating the need to use additional resources to deposit it manually or mechanically. | - The introduction of water in the backfill to the mine is a problem if drainage is done by pumping. |
| - The surface left by the cemented hydraulic fill means that there is no wear on the equipment's tires that can transit through the work in the fill. | - When using material with high pyrite content, when these sulfides are oxidized, an exothermic reaction occurs, which raises the temperature and produces sulfur anhydride. |
| - The cemented hydraulic fill allows to increase the efficiency and productivity in the pits due to the decrease in the consumption of wood and the reduction of the mining cost due to the versatility it offers. | - In the drainage water of the backfill it always carries a certain quantity of fines and portions of the cement binder which are deposited in the lower levels of the filled works. |

| Cemented Paste Backfill | |
| --- | --- |
| **Advantages** | **Disadvantages** |
| - Backfill material with low segregation. | - High investment cost (dosing plant + mixing system + pumping system + pipeline network) |
| - It can be pumped for great distances with a suitable high-pressure pump (greater than 2000 m). | - Presence of fines in tailings produces complex rheology for pipeline transport. |
| - Additives can be used to keep the mixture fresh and with adequate rheology. | - In some cases, the use of high amounts of cement and additives is required. |

As a consequence of tailings storage facility failures around the world, the international community, institutions, and global groups such as the ICMM (International Council on Mining and Metals), the UN (United Nations) environmental program, and PRI (Principles for Responsible Investments) have developed Global Industry Standard on Tailings Management (GISTM), launched in August 2020, to regulate the operation of tailings storage facilities throughout their entire life cycle, including closure and post-closure (perpetuity),

with the goal of zero damage to people and the environment and zero tolerance for human deaths. Unfortunately, key aspects related to the management of mine tailings have not been considered in detail, including (i) the promotion of alternative techniques against conventional methods for mine tailings management, (ii) the definition of backfilling with mine tailings of underground mines, (iii) the generation, prevention, and control of acid rock drainage, (iv) the potential for metal leaching, and (v) toxicity issues. There is a lack of definition and discussion of these issues in the Global Industry Standard on Tailings Management (GISTM). It is urgent to carry out comprehensive, multidisciplinary, transdisciplinary, and holistic governance of mine tailings that incorporates these issues into the Global Standard of Tailings Management for the Mining Industry in order to promote sustainable and responsible mining.

Unfortunately, there is no detailed policy document that regulates the scope of use, opportunities, and restrictions for mining operations that consider the use of mine tailings as backfill for underground mines in Peru. In this context, local mining companies consider past practical experiences in underground mine backfilling and additionally hire specialized engineering companies who develop engineering projects considering the highest standards, experiences, and good practices carried out in other mining countries such as Australia, South Africa, and Canada. Considering this background, as a finding of this research, it is possible to mention that the accumulated experience in Peru since 1937, considering the first underground mine that used hydraulic fill in Cerro de Pasco mine, is significant and valuable. This is mainly due to the success achieved to date, considering the complex site-specific conditions of the mines in Peru. In this sense, local authorities should publish a practical methodological and normative guide/policy document for free public access that indicates restrictions, opportunities, lessons learned, and scope of use for promoting the implementation of backfilling with mine tailings in underground mines.

Considering the highly aggressive geochemical characteristics of the mineralogy of the mining tailings in Peru with the presence of pyrite and its negative effects when interacting surface with oxygen and water by producing metal leaching and the generation of acid rock drainage (ARD), it is mitigated by considering the use of mine tailings in backfilling of underground mines [13,30]. Implementing this technique has mitigated the environmental impact on surface and groundwater in the hydrographic basins where the communities neighboring the mining operations are located [65]. This has benefited the agricultural and livestock activities carried out by the communities through the cultivation and breeding of animals such as the high Andean camelids [16].

Another aspect improved with the implementation of this technique is the elimination of mine closure works on surface terrain, as is necessary in the case of conventional tailings storage facilities built on the surface, and the elimination of a perpetual acid rock drainage (ARD) treatment system. This reduces the capital and operating costs of mine closure measures and also generates sustainable environmental conditions over time for all stakeholders in the territory [21].

It is important to mention that storing mine tailings inside underground mines eliminates the socio-environmental problems associated with the emission of particulate matter that is caused by conventional tailings storage facilities. This brings many benefits for the quality of life and health of neighboring communities to neighboring operations, reducing the risk of bioaccumulation of heavy metals in children and adults, as well as improving the quality and productivity of agricultural and livestock activities related to crops and animal husbandry. For example, the risk of animals accidentally entering the mine tailings storage area to drink water is eliminated [13,16].

In addition, the risks of failure of tailings storage facility dams due to seismic and extreme weather events have been considerably reduced, thus avoiding the spillage of mine tailings into the environment [66,67].

An underground mine tailings storage option has a considerable advantage, from an environmental impact perspective, in that it can substantially meet best industry standards and possibly best practices. Since acid rock drainage (ARD) generation is controlled in

the period from mining to closure, concerns about contamination from mine waste are low, and this would be considered a "best" practice, a major advantage for the approval of environmental impact assessment (EIA) studies and obtaining a social and environmental license to operate [68].

Finally, considering all these aspects, it is possible to mention that the filling of underground mines with the use of mine tailings is a sustainable green mining solution under the concept of a circular economy where mining waste is reused in the mining process, reducing socio-environmental impacts compared to conventional tailings storage methods [69,70].

## 6. Conclusions

Considering the socio-environmental restrictions imposed by local communities due to water, soil, and air pollution, as well as the complex topographical conditions with steep slopes and extreme weather conditions (liquid and solid precipitation during the wet season), storing mine tailings on the surface becomes unfeasible in some cases. This, in turn, makes filling underground mines with mine tailings an attractive alternative to ensure mining operations.

The filling of underground mines with mine tailings allows the return of mining waste material to its origin, considerably reducing the amount of mine tailings deposited on the surface, thus decreasing the number of mining wastes and mitigating the environmental impacts of the territory. This alternative mine tailings storage solution complies with the sustainability principles related to waste management on the 3R paradigm: reduce, reuse, and recycle, promoting responsible mining activity with communities and the environment.

This is how this green mining solution considers the concepts of circular economy, reusing the mining waste obtained from the metallurgical process and reintegrating this form of matter with no economic value to its place of origin, which is the underground mine. In this way, negative socio-environmental impacts on neighboring communities and the environment are considerably reduced, both at the spatial and temporal scales of land use. Storing mine tailings underground reduces the impact on the surface. This practice is more environmentally friendly as it eliminates the need to use surface soils for mining waste disposal, thus avoiding problems related to dust generation, visual impact, and surface/underground water contamination. Additionally, it reduces the risk of flood and earthquake events associated with potential structural failure of tailings storage facility dams.

Local authorities of Peru should prepare a normative document and methodological guide to promote the implementation of the filling of underground mines with mine tailings, sharing its mining experience of decades in complex site-specific conditions of the Andean region and thus promote a green mining solution that can be shared with the rest of the world.

Finally, it is possible to mention that the underground mine fill with the use of mine tailings is a green mining solution that allows for reducing socio-environmental impacts and improving the sustainability of the stakeholders in the territory, mainly due to: (i) mitigation of drainage rock acid (ARD), (ii) elimination of the risk of failure of tailings containment dams in the face of seismic hazard, (iii) reduction of potential seepage into watershed aquifers, (iv) elimination of particulate material emissions, (v) reduction of the visual impact of large dams, (vi) mitigation of the degradation of terrestrial habitats, (vii) promotion of alternative land uses on the surface. This responsible and sustainable green mining solution will become more and more popular over time and will be used in many underground mining projects around the world in order to guarantee the continuity of the mining business, ensure good relations with communities and preserve local ecosystems.

**Author Contributions:** Conceptualization, C.C. and A.M.; formal analysis, C.C.; investigation, C.C.; resources, A.M.; writing—original draft preparation, C.C.; writing—review and editing, C.C. and A.M.; visualization, C.C.; supervision, A.M. All authors have read and agreed to the published version of the manuscript.



**Funding:** The research is funded by the Research Department of Catholic University of Temuco, Chile and by the Research Department of Universidad Privada del Norte, Peru.

**Data Availability Statement:** The data presented in this study are available on request from the corresponding author.

**Conflicts of Interest:** The authors declare no conflict of interest.

**Abbreviations**

| | |
|---|---|
| TSF | Tailings Storage Facility |
| UN | United Nations |
| GISTM | Global Industry Standard on Tailings Management |
| ICMM | International Council on Mining and Metals |
| PRI | Principles for Responsible Investment |
| ESG | Environmental, Social and Governance |
| 3R | Reduce, Reuse and Recycle |
| BATs | Best Available Technologies |
| CTD | Conventional Tailings Disposal |
| TTD | Thickened Tailings Disposal |
| PTD | Paste Tailings Disposal |
| FTD | Filtered Tailings Disposal |
| UMB | Underground Mine Backfilling |
| HF | Hydraulic Fill |
| CHF | Cemented Hydraulic Fill |
| CPB | Cemented Paste Backfill |
| EIA | Environmental Impact Assessment |
| Cw | Slurry tailings solids content by weight |
| mtpd | Metric tonnes per day |
| PD Pumps | Positive Displacement Pumps |
| masl | Meters above sea level |
| ARD | Acid Rock Drainage |

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
