# Peer review of "Experiences of Underground Mine Backfilling Using Mine Tailings Developed in the Andean Region of Peru: A Green Mining Solution to Reduce Socio-Environmental Impacts"

_sustainability, doi:10.3390/su151712912_

Round 1

Reviewer 1 Report

(1) Some references are incorrectly formatted, for example, the page number of reference [1] is incorrect. Please carefully check other reference formats.

(2) It is recommended to increase the policy documents on tailings treatment issued by the local government.

(3) Section 4 only divided 4.1 subheadings unreasonable, it is recommended to delete.

  • The language can be better polished.

Reviewer 2 Report

The manuscript is novel only in that it submits information relating, in small part, to the Peruvian underground mining process with some limited case study data.The scope is generally wide ranging on the topic of mining backfill processes, but not unique in its generic descriptions of mine backfill types.

Thew writing quality is fair, with too many cases of run-on sentence structures and repeated descriptions of techniques that make easy reading (in English) of the text only fair.  Descriptions of backfill manufacturing technologies are not unique to Peru and have been described in better fashion by many other authors.   The manuscript could be enhanced by more concise editing, and may have some technical problems as noted within the edited text (ie.- use of positive displacement pumping systems to provide underground transport of hydraulic fills from surface to underground stopes;  this is a delivery technique required for paste backfill delivery but not for hydraulic fill materials).

The data relating to Peruvian mining operations is very useful for the subject statement outlined but much of the included information and schematics may not be based upon Peru mining activity - if it is based on local activity then it should be better identified.

Overall the paper is useful but could benefit from substantial editing and shortening.   The English language text is understandable, but numerous editing changes (partially indicated by text editing made on the attached document) will be required.

As above

Reviewer 3 Report

I would like to congratulate the authors for the originality and structure of the article. I believe it has great potential and room for further development. However, I have identified a few details that I believe should be corrected:

First, in the introduction, the phrase "In Peru and in the rest of the world, the limitations in the supply of water for mining processes and the sustainable disposal of mine tailings have become central issues for the viability of this type of project and obtaining the social and environmental license to operate in the territory" could be rephrased for better clarity and flow.

This text demonstrates an advanced level of English, although it contains some grammatical and punctuation errors. The structure is coherent, establishing the importance of issues related to the disposal of mine tailings and obtaining social and environmental licenses. However, it requires a more detailed revision to correct the grammatical errors and improve the fluency of the text.

Regarding the specific errors, the following have been identified:

  • In line 29, "limitations in the supply of water for mining" should be "limitations on the supply of water for mining."
  • In line 39, "to the investment of mining projects such as ICMM, UN, and PRI are promoting" should be "to organizations involved in mining investment projects, such as ICMM, UN, and PRI, who are promoting."
  • In line 47, "currently mandatory in Peru" should be "which are currently mandatory in Peru."
  • In line 51, "the so-called best available technologies (BATs), which are:" should be "the so-called best available technologies (BATs), including:"
  • In line 106, "together with the scope, precautions, and application limitations of this solution are discussed" should be "together with a discussion of the scope, precautions, and application limitations of this solution."
  • In line 109, "improving the sustainability of the actors in the territory" should be "improving the sustainability of stakeholders in the area."

In the conclusion I have detected some mistakes which I think must be corrected: 

Error 1: Lack of clarity in presenting the environmental and topographical conditions.

Original: Considering socio-environmental restrictions of local communities who claim free 613 pollution for environmental conditions of water, soil, and air, together with the complex 614 topographical conditions with steep terrain on steep slopes and also with extreme weather 615 conditions with liquid precipitation and solid in the wet season, make in some cases the 616 storage of mine tailings on the surface unfeasible, this allows the filling of underground 617 mines with the use of mine tailings an attractive alternative to ensure the mining opera- 618 tion.

Revised: Considering the socio-environmental restrictions imposed by local communities due to water, soil, and air pollution, as well as the complex topographical conditions with steep slopes and extreme weather conditions, such as liquid and solid precipitation during the wet season, storing mine tailings on the surface becomes unfeasible in some cases. This makes the filling of underground mines with mine tailings an attractive alternative to ensure mining operations.

Error 2: Inappropriate use of the word "claim" and lack of coherence in the writing.

Original: Considering socio-environmental restrictions of local communities who claim free 613 pollution for environmental conditions of water, soil, and air, together with the complex 614 topographical conditions with steep terrain on steep slopes and also with extreme weather 615 conditions with liquid precipitation and solid in the wet season, make in some cases the 616 storage of mine tailings on the surface unfeasible, this allows the filling of underground 617 mines with the use of mine tailings an attractive alternative to ensure the mining opera- 618 tion.

Revised: Taking into account the socio-environmental restrictions imposed by local communities due to water, soil, and air pollution, as well as the complex topographical conditions with steep slopes and extreme weather conditions, such as liquid and solid precipitation during the wet season, storing mine tailings on the surface becomes unfeasible in some cases. This makes filling underground mines with mine tailings an attractive alternative to ensure mining operations.

Error 3: Long sentences and lack of content organization.

Original: Considering socio-environmental restrictions of local communities who claim free 613 pollution for environmental conditions of water, soil, and air, together with the complex 614 topographical conditions with steep terrain on steep slopes and also with extreme weather 615 conditions with liquid precipitation and solid in the wet season, make in some cases the 616 storage of mine tailings on the surface unfeasible, this allows the filling of underground 617 mines with the use of mine tailings an attractive alternative to ensure the mining opera- 618 tion.

Revised: Considering the socio-environmental restrictions imposed by local communities due to water, soil, and air pollution, as well as the complex topographical conditions with steep slopes and extreme weather conditions (liquid and solid precipitation during the wet season), storing mine tailings on the surface becomes unfeasible in some cases. This, in turn, makes filling underground mines with mine tailings an attractive alternative to ensure mining operations.

Error 4: Lack of evidence and specific data to support the claims.

Original: Mine tailings are stored underground and reduce the impact on the surface. This 624 condition is more friendly to the environment since surface soils do not need to be used 625 as mining waste disposal areas, since problems associated with dust generation, visual 626 impact, and contamination of surface/underground watercourses are avoided, and reduc- 627 tion of flood risks associated with possible structural failure of tailings storage facility 628 dams. 629

Revised: Storing mine tailings underground reduces the impact on the surface. This practice is more environmentally friendly as it eliminates the need to use surface soils for mining waste disposal, thus avoiding problems related to dust generation, visual impact, and surface/underground water contamination. Additionally, it reduces the risk of floods associated with potential structural failure of tailings storage facility dams.

Reviewer 4 Report

The submitted article for review is of an overview type and is devoted to summarizing the experience of goaf backfilling in the mines of Peru.

The article may be useful for specialists in the field of backfilling. However, after carefully reading the research material, I had comments and recommendations solely to improve the quality of the article.

1. Subsection 2.1 I would suggest titled "Characteristics of Paste Backfilling Technologies in Peruvian Mine Conditions" (or a similar title).

2. Figure 1 has no value, since it shows the entrance to the mine (probably an adit). I recommend removing it.

3. It would be necessary to indicate in the text and in the caption to Figure 2 which mine is being considered.

4. In my opinion, figure 3 is redundant. It doesn't make any sense. It is clear that if pipeline transport is used for paste backfilling, then the pipeline is laid in a mine working.

5. In Figure 4 and 6, I would recommend making symbols (1, 2, 3, 4…). Indicate the stope face, wells, underground mine workings. Specify the height of the chamber, the sublevel drifts.

6. What does the meaning of S1, S2… in figure 5 mean? It's probably a stopes.

7. The authors analyze different types of backfill. But I have not seen answers to the question, how is a particular type of backfill selected in the conditions of mines in Peru? By what criteria. Analyze it.

8. In my opinion, there is a lot of well-known information on backfill technology in the article, especially in subsections 2.2, 2.3, 2.4, 3. Please pay more attention to the experience and peculiarities of mines in Peru.

9. In my opinion, it is incorrect to provide links to articles in the "Discussion" section. This is useful when the article is experimental in nature and where you are comparing your results with previous ones. References are given here as a statement of fact. Summarize, discuss the advantages, disadvantages and prospects of backfilling methods in Peru. What you analyzed in the article.

10. Findings are generally known for backfill technology. Provide a summary of your review in the article. What backfilling methods prevail in mines? Why exactly them? What is it connected with? Pay attention to the experience in Peru. After all, this is the practical value of your work.

11. I would recommend adding a paragraph in the introduction to say that: many mines around the world use different backfill technologies such as rockfill […], hydraulic fill […], paste backfill […]. It was the presence of tailings near the mines as backfill material and their danger to the environment that contributed to the development of backfilling technologies in the mines of Peru - (types used in Peru). This will show the importance of backfill technology in the world and show how it is different for mines in Peru.

I recommend in the list of references for “rockfill” to mark a mine in Ukraine, where uncemented blast furnace slag based rockfill is used:

Kuzmenko, O., Petlyovanyy, M., & Heylo, A. (2014). Application of fine-grained binding materials in technology of hardening backfill construction. Progressive Technologies of Coal, Coalbed Methane, and Ores Mining, 465-469. https://doi.org/10.1201/b17547-79

12. Pay attention to the design of the list of references for the article. I don't see doi to articles. Look for the latest published journal articles for literature design.

Round 2

Reviewer 4 Report

The authors did a good job updating the article.

I saw clarification and reasonable answers to some of the discussion questions.

Still, in my opinion, the photo in figure 1 does not make much sense and is superfluous.

This decision is up to the editor.

Nevertheless, the article is interesting and will be useful for expanding knowledge in the field of backfilling.

I wish the authors scientific success.

Sincerely, Reviewer